# The relationship between executive functions and fluid intelligence in multiple sclerosis

**Belén Goitia**[1,2,3], **Diana Bruno**[1], **Sofía Abrevaya**[1,3], **Lucas Sedeño**[1,3], **Agustín Ibáñez**[1,3,4,5,6], **Facundo Manes**[1,3], **Mariano Sigman**[2,3], **Vladimiro Sinay**[1], **Teresa Torralva**[1], **John Duncan**[7,8], **María Roca**[1,3]*

**1** Institute of Translational and Cognitive Neuroscience (INCyT), INECO Foundation, Favaloro University, Buenos Aires, Argentina, **2** Laboratory of Neuroscience, Torcuato Di Tella University, Buenos Aires, Argentina, **3** National Scientific and Technical Research Council (CONICET), Buenos Aires, Argentina, **4** Center for Social and Cognitive Neuroscience (CSCN), School of Psychology, Universidad Adolfo Ibáñez, Santiago, Chile, **5** Universidad Autónoma del Caribe, Barranquilla, Colombia, **6** Centre of Excellence in Cognition and its Disorders, Australian Research Council (ACR), Sydney, Australia, **7** Medical Research Council (MRC) Cognition and Brain Sciences Unit, Cambridge, United Kingdom, **8** Department of Experimental Psychology, University of Oxford, Oxford, United Kingdom

* mroca@ineco.org.ar

**Data Availability Statement:** Data cannot be shared publicly because public availability would compromise patient confidentiality or participant privacy. This is due to the small sample size in our study (less than 100 individuals), the fact that the

## Abstract

### Background & objective

Deficits in cognitive functions dependent upon the integrity of the prefrontal cortex have been described in Multiple Sclerosis (MS). In a series of studies we have shown that fluid intelligence (*g*) is a substantial contributor to frontal deficits and that, for some classical "executive" tasks, frontal deficits were entirely explained by *g*. However, for another group of frontal tasks deficits remained once *g* was introduced as a covariate. This second set of tests included multitasking and theory of mind tasks. In the present study, we aimed at determining the role of fluid intelligence in frontal deficits seen in patients with MS.

### Methods

A group of patients with Relapsing Remitting MS (n = 36) and a group of control subjects (n = 42) were assessed with a battery of classical executive tests (which included the Wisconsin Card Sorting Test, Verbal Fluency, and Trail Making Test B), a multitasking test, a theory of mind test and a fluid intelligence test.

### Results

MS patients showed significant deficits in the fluid intelligence task. We found differences between patients and control subjects in all tests except for the multitasking test. The differences in the classical executive tests became non-significant once fluid intelligence was introduced as a covariate, but differences in theory of mind remained.

place of treatment can be easily inferred from our affiliations, and that we have taken sex and age into account in our analyses. This has led the ethics committee at INECO to advice against making our data public. Should other researchers need access to the data used for this study, they can send a request to comite@ineco.org.ar.

**Funding:** The author(s) received no specific funding for this work.

**Competing interests:** The authors have declared that no competing interests exist.

## Conclusions

The present results suggest that fluid intelligence can be affected in MS and that this impairment can play a role in the executive deficits described in MS.

## Introduction

Multiple Sclerosis (MS) is a neurological disease characterized by demyelination and axonal loss, which results in disruption of neuronal communication in the Central Nervous System. Of the four MS subtypes defined by their clinical course, Relapsing Remitting MS (RRMS) is the most common, with 85% of MS patients falling into this category [1].

Even though a few decades ago cognitive deficits were considered uncommon among MS patients, it is now estimated that 43–65% of MS patients suffer from cognitive impairment [2–5]. Deficits have been reported in complex attention, efficiency of information processing, executive functioning, processing speed, and long-term memory and are known to impact daily living abilities [6]. Such deficits have been related to disruptions in white matter tracts, particularly in frontal-subcortical networks [7–12], but also to depressive symptoms [13,14] or high fatigue levels [13,15], though some studies argue against the latter relationship [16–18].

A wide variety of cerebral structures and mechanisms have been linked to MS cognitive deficits. Reports include cortical volume loss [19,20], reduced cerebral blood flow in the frontal lobes [21], volume loss in deep grey matter structures, particularly the thalamus, hippocampus and putamen [22–27], and focal white matter and grey matter damage [27,28].

Executive impairment in MS has been related to damage in fronto-subcortical tracts. In clinical neuropsychological studies, the prefrontal cortex (PFC) is believed to support "executive functions". Executive functions are understood as processes that organize behaviour executing cognitive control of lower order functions. Within executive functions, many processes can be identified, such as planning, flexibility, switching and inhibition. Both in clinical and in experimental fields multiple tests have been used to measure each function, including the Wisconsin Card Sorting Test (WCST), Verbal Fluency and the Trail Making Test B as the most widely used in clinical and experimental neuropsychology. Besides these classical executive tests, damage to PFC has also been associated with deficits in social cognition and multitasking.

Parallel to specific frontal functions supposedly associated with specific PFC regions, experimental neuroscience has linked the frontal lobe to the concept of "general intelligence" or "*g*", introduced by Spearman (1904, 1927) to account for universal positive correlations between different cognitive tests. More generally, factor *g* has been linked to a multiple-demand (MD) [29, 30] system—including regions of the lateral frontal surface, the middle frontal gyrus, the premotor cortex, the anterior insula, and the dorsomedial frontal cortex, accompanied by a further region in lateral parietal cortex [31]—that is active during many different kinds of cognitive tasks.

The importance of the frontal lobe in fluid intelligence made us question the nature of the relationship between *g* and the above frontal tasks. If *g* is positively correlated with all tasks, then *g* deficits in frontal patients may explain bad performance in frontal tasks. We wanted to know how well deficits in tasks associated with PFC were explained by a fluid intelligence loss. In this regard, an early paper from our group [32] showed that fluid intelligence is a substantial contributor to cognitive deficits in frontal patients; however, this relationship is not simple.

While for classical executive tasks once fluid intelligence is partialled out, no deficits remained, for a second set of tasks—mainly tasks of multitasking and social cognition—deficits remained. We showed similar results in many conditions, such as frontal lobe lesions [32], Parkinson's disease [33], Frontotemporal Dementia [34], Schizophrenia [35], and Bipolar Disorder [36].

Our results have had strong implications for the use and interpretation of widely used tests such as the Wisconsin Card Sorting Test, Verbal Fluency and Trail Making Test B. According to our results, deficits measured in such tests were not reflecting problems specific to their particular cognitive content—such as flexibility, energisation or switching—but instead they might be reflecting a much broader cognitive loss. On the other hand, deficits in social cognition and multitasking seemed not to be explained in this way.

Our aim in this paper is to test whether frontal deficits measured by frontal tests in MS patients can be accounted for by a deficit in *g*. Similar to our previous work, we introduce *g* as a covariate to see to what extent frontal deficits remain. If the same pattern that we have found in the past is repeated, then we expect that impairments in classical executive tests will be completely explained by reduced *g*, while deficits in other tests—here including multitasking and social cognition—will remain. In this paper we also searched for a link between MS cognitive deficits and the MD system [37] asking if volume loss in the MD system, as measured with MRI, correlates with the general intelligence factor in this population.

## Methods

### Participants

Permission for the study was obtained from the local research ethics committee (INECO Foundation) and all participants gave their signed informed consent prior to inclusion, according to the Declaration of Helsinki of 1975, as revised in 2008. Thirty-six patients diagnosed with Relapsing Remitting MS (30 women, 6 men), fulfilling Poser and McDonald criteria [38,39] and referred to our MS clinic for routine follow-up, underwent neuropsychological and MRI evaluation for the present study. All had mild clinical disability [Expanded Disability Status Scale (EDSS) <2], without visual deficit or upper limb impairment potentially affecting neuropsychological test performance or history of alcohol or drug abuse, major psychiatric disorder, head trauma or other neurological disorder or systemic illness. All tests were performed at least 90 days after the most recent relapse episode, and with all patients off steroid treatment for at least three months. Mean age was 39.2 ± 10.2 years (range 21–64 years) and mean disease duration 9.3 ± 7.3 years (range 1–35 years). Physical disability was assessed using EDSS [40] and MS Functional Composite (MSFC) score [41]. Forty-two subjects matched for age, gender (29 women, 13 men) and educational level recruited from a local volunteer group served as controls. Participants were included in the control group if they reported no history of neurological or psychiatric disorders, including traumatic brain injury or substance abuse. To control for mood symptoms and fatigue in the patients with MS, we assessed them with the Beck Depression Inventory [42] and the Modified Fatigue Impact Scale [43].

All participants in the study (patients and controls) were examined to ensure they had no comorbidity with other psychiatric or neurological disorders.

### Neuropsychological assessment

To estimate pre-morbid intelligence we used the Word Accentuation Test-Buenos Aires edition [44]. This test, similar to the National Adult Reading Test [45], measures the ability to read 51 irregularly stressed Spanish words.

**Fluid intelligence (*g*).** Matrix Reasoning is a subtest of WAIS-III [46] that gives a measure of fluid reasoning. In this test the subject is presented with an incomplete pattern and given 5 options for completing it properly. Each correct answer gives a score of 1 point (0 for incorrect answers). The test is interrupted after 4 consecutive incorrect answers. The maximum gross score is 26 points. We have taken this as our measure of *g*.

## Classical executive tests

**Wisconsin card sorting test (WCST)** [47]**.** For the WCST we used Nelson's modified version of the standard procedure. Cards varying on three basic features—colour, shape and number of items—must be sorted according to each feature in turn. The participant's first sorting choice becomes the correct feature, and once a criterion of six consecutive correct sorts is achieved, the subject is told that the rules have changed, and cards must be sorted according to a new feature. After all three features have been used as sorting criteria, subjects must cycle through them again in the same order as they did before. Each time the feature is changed, the next must be discovered by trial and error. Score was total number of categories achieved. Data were available for 35 patients.

**Verbal fluency** [48]**.** In verbal fluency tasks, the subject generates as many items as possible from a given category. We used the standard Argentinian phonemic version, asking subjects to generate words beginning with the letter P in a one-minute block. Score was the total number of correct words generated. Data were available for 36 patients.

**Trail making test B (TMTB)** [49]**.** In this test the subject is required to draw lines sequentially connecting 13 numbers and 12 letters distributed on a sheet of paper. Letters and numbers are encircled and must be connected alternately (e.g., 1, A, 2, B, 3, C, etc.). Score was the total time (s) required to complete the task, given a negative sign so that higher scores meant better performance. Data were available for 35 patients.

## Multitasking and social cognition tests

**Hotel task** [50,51]**.** The hotel task, originally designed by Manly in 2002, has been used as an ecological assessment tool to evaluate goal management and multitasking abilities in different neurological and psychiatric conditions. The test requires planning, problem solving abilities, prospective memory, organizing and monitoring behavior. The task comprised five primary activities related to running a hotel (compiling bills, sorting coins for a charity collection, looking up telephone numbers, sorting conference labels, proofreading). The materials needed to perform these activities were arranged on a desk, along with a clock that could be consulted by removing and then replacing a cover. Subjects were told to try at least some of all five activities during a 15 min period, so that, at the end of this period, they would be able to give an estimate of how long each task would take to complete. It was explained that time was not enough to actually complete the tasks; the goal instead was to ensure that every task was sampled. Subjects were also asked to remember to open and close the hotel garage doors at specified times (open at 6 min, close at 12 min), using an electronic button. Of the several scores possible for this task, we used time allocation: for each primary task we assumed an optimal allocation of 3 min, and measured the summed total deviation (in seconds) from this optimum. Total deviation was given a negative sign so that higher scores meant better performance. Data were available for 31 patients.

**Faux pas** [52]**.** In each trial of this test, the subject was read a short, one paragraph story. To reduce working memory load, a written version of the story was also placed at all times in front of the subject and available to re-read as many times as needed. In 10 stories there was a faux pas, involving one person unintentionally saying something hurtful or insulting to

another. In the remaining 10 stories there were no faux pas. After each story, the subject was asked whether something inappropriate was said and if so, why it was inappropriate. If the answer was incorrect, an additional memory question was asked to check that basic facts of the story were retained; if they were not, the story was re-examined and all questions repeated. The score was 1 point for each faux pas correctly identified, or non-faux pas correctly rejected.

## Statistical analysis of neuropsychological data

All statistical analyses regarding neuropsychological results were performed with IBM SPSS® Statistics 20. Groups were compared through Student's t-tests for the following variables: age, education years, WAT-BA, WCST, Verbal Fluency, TMTB, Hotel task, Faux Pas. Groups were compared again, this time taking Matrix Reasoning as a covariate through an ANCOVA, for the following variables: WCST, Verbal Fluency, TMTB, Hotel task, Faux Pas.

## Image acquisition

We obtained MRI recordings from 28 multiple sclerosis patients and 29 controls. Subjects were scanned in a 1.5 T Philips Intera scanner with a standard head coil. We used a T1-weighted spin echo sequence that covered the whole brain (matrix size = $256 \times 240 \times 120$, 1 mm isotropic; TR = 7489 ms; TE = 3420 ms; flip angle = 8˚).

## Grey-matter analysis

A grey-matter analysis was performed to establish the participants' grey matter volume. Data were preprocessed on the DARTEL Toolbox following validated procedures [53–57] and using Statistical Parametric Mapping software (SPM12) (http://www.fil.ion.ucl.ac.uk/spm/software/spm12/). Images were segmented into grey matter, white matter, and cerebrospinal fluid volumes. Next, images were smoothed with a 12 mm full-width half-maximum kernel as proposed in other reports [53,58] and normalized to MNI space. To test whether the performances on the Matrix Reasoning test (*g*) and the Faux Pas test (the latter being expected not to be related to *g*) were associated with the MD system, we restricted our analysis using a mask of the main areas of this network (http://imaging.mrc-cbu.cam.ac.uk/imaging/MDsystem) [59], to extract the grey matter volume for each participant. The preprocessed images were used to extract the grey matter volume (in ml) of this mask for each participant with a toolbox that runs in the MATLAB environment (The MathWorks, Inc., Natick, Massachusetts; Ged Ridgway, http://www.cs.ucl.ac.uk/staff/g.ridgway/vbm/get_totals.m), and has been used in previous studies [60–64]. Once the values were obtained, we performed non-parametric Spearman correlation tests between either Matrix Reasoning Raw scores and Faux Pas and the MD grey matter volume with STATISTICA version 10 (StatSoft, Inc., 2011, www.statsoft.com.). Considering the size of our experimental samples (<30), we first applied a previously used strategy [65–69] of combining samples to add greater variance, thereby increasing the statistical power of the study to detect associations. This approach provides knowledge regarding a general association between brain markers and each cognitive task. Complementary to this analysis, non-parametric Spearman correlation tests were also calculated for each separate group to evaluate the critical brain areas for each group. In addition, we selected two control networks, the somatosensory and default mode network (based on the AAL atlas [70], as done in previous research [71], to evaluate the specificity of association with the MD system. Bonferroni correction was applied to control for the multiple comparison problem for each group and index associations (*p*-value corrected < .01).

## Results

Clinical and demographic data for all participants are shown in Table 1. Neuropsychological results are shown in Table 2. Patients and controls were compared for WCST, Verbal Fluency and TMTB using two-tailed t-tests. The MS group showed statistically significant impairments on all three tests from the classical executive battery, as had been expected: WCST, $t(75) = -2.8$, $p = 0.007$; Verbal Fluency, $t(75) = -2.1$, $p = 0.041$; TMTB, $t(75) = 4.13$, $p < 0.001$. The MS group was also significantly impaired in the Faux Pas task ($t(76) = -5.2$, $p < 0.001$). Unexpectedly, no differences were found between groups in the Hotel task ($t(70) = 1.5$, $p = 0.148$).

All three classical executive tests showed correlation with Matrix Reasoning. The average within-group correlations with Matrix Reasoning, after combining data from patients and controls, were r = 0.322 for WCST, r = 0.228 for Verbal Fluency, and r = 0.575 for TMTB (Table 2). Higher Matrix Reasoning values were associated with better performance in all three executive tasks, as shown in the scatter plots (Fig 1); beyond this linear regression, there was no apparent additional group effect. For a better assessment, the two groups were again compared, but Matrix Reasoning was taken as a covariate (ANCOVA). After this, significant differences between patients and controls in all three classical executive tasks were no longer found: for WCST, $F = 1.784$, $p = 0.186$; for Verbal Fluency, $F = 0.767$, $p = 0.384$; and for TMTB, $F = 3.593$, $p = 0.062$ (Table 2). This suggests that, for the classical executive tasks, frontal deficits were largely explained by fluid intelligence. On the contrary, for the Faux Pas task significant group differences remained after including Matrix Reasoning as a covariate (ANCOVA): $F = 18.300$, $p < 0.001$ (Table 2).

Since it has been suggested that fatigue and depression in MS patients might explain their cognitive deficits [13,72], we also ran ANCOVAs taking either fatigue or depression as the covariate instead of Matrix Reasoning. In the case of fatigue, the significant differences between groups for all tasks remained, while in the case of depression significant differences between groups remained for all tests except for Verbal fluency (Table 2).

### Grey matter analysis

When patients and controls were combined, we found a significant correlation between the MD system grey matter volume and Matrix Reasoning Raw scores. Non-significant correlations were found between MD system grey matter volume and the Faux Pas total score. When correlations were calculated separately for the control and the patient groups, the correlation between MD grey matter and Matrix Reasoning Raw scores was significant only in the control group (Table 3). In the patient group no significant correlations were found in either of the comparisons. The DMN mask showed the same results; however, the somatosensory mask only presented significant correlation in the analysis combining control and patient's samples. Drawing strong conclusions is hard, however, as grey matter volumes in the 3 networks were themselves strongly correlated (Table 4). This means we had little power to separate their effects.

**Table 1. Clinical and demographic data for MS patients and controls.**

| | MS | | Controls | | *p* (two-tailed Student's *t*-test) |
|---|---|---|---|---|---|
| | Mean | S.D. | Mean | S.D. | |
| Age (years) | 39.2 | 10.2 | 37.1 | 10.7 | 0.376 |
| Education (years) | 16.6 | 3.3 | 16.8 | 2.7 | 0.771 |
| WAT-BA | 41.9 | 5.5 | 43.6 | 5.0 | 0.155 |

**Table 2. Patient and control scores, average within-group correlation with *g*, and significance of group differences for each task.**

| | MS | | Controls | | Patients *vs.* controls (*p*) | Average within-group correlations with Matrix Reasoning | Patients *vs.* controls after adjustment for Matrix Reasoning (*p*) | Patients *vs.* controls after adjustment for fatigue (*p*) | Patients *vs.* controls after adjustment for depression (*p*) |
|---|---|---|---|---|---|---|---|---|---|
| | Mean | S.D. | Mean | S.D. | | | | | |
| *G* | 9.2 | 3.2 | 12.0 | 2.4 | < .001 | - | - | | |
| Fatigue | 39 | 20.0 | 19.1 | 15.3 | < .001 | | | | |
| WCST | 5.1 | 1.5 | 5.8 | 0.6 | .007 | .322 | .186 | .013 | .028 |
| Verbal Fluency | 15.4 | 5.1 | 18.9 | 5.5 | .041 | .228 | .384 | .040 | .166 |
| TMTB | -113.4 | 66.8 | -66.7 | 27.9 | < .001 | .575 | .062 | .041 | .010 |
| Hotel task | -466.2 | 209.0 | -400.4 | 172.3 | .148 | | | | |
| Faux Pas | 17.5 | 1.8 | 19.4 | 0.9 | < .001 | .103 | < .001 | < .001 | < .001 |

## Discussion

Many frontal deficits have been described as a part of the cognitive profile in patients with MS. Impaired performance has been described in classic executive tests and in tests of multitasking and social cognition. Since the frontal lobe has also been related to fluid intelligence, we asked how far frontal deficits can be explained by a general fluid intelligence loss. We found that for the classical executive tests included in this investigation (WCST, Verbal Fluency and TMTB) the differences between MS patients and controls became non-significant once fluid intelligence was introduced as a covariate. On the contrary, differences on the Faux Pas test, which gives a measure of social cognition (Theory of Mind), remained significant after adjustment for fluid intelligence.

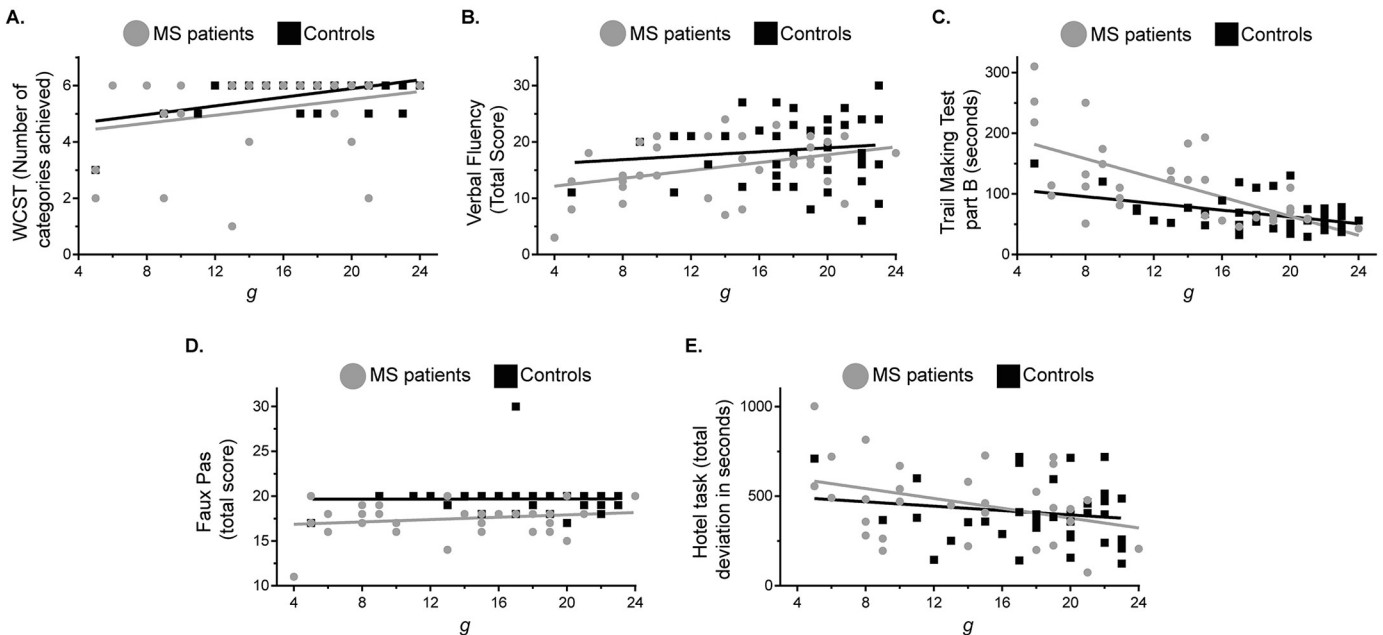

**Fig 1. Performance in classical executive tests.** Scatter plots relating performance in (A) the Wisconsin Card Sorting Test (WCST), (B) Verbal Fluency, (C) Trail Making Test part B (TMTB), (D) Faux Pas, and (E) Hotel task to Matrix Reasoning for patients with multiples sclerosis (circles) and controls (squares). Regression lines (grey for multiple sclerosis and black for controls) reflect the correlation values for each group.

**Table 3. Correlation between grey matter and test scores.**

| Mask | Group | Matrix Reasoning (*g*) | | Wisconsin Card Sorting Test (WCST) | | Verbal Fluency | | Trail Making Test B | | Hotel task | | Faux Pas | |
|---|---|---|---|---|---|---|---|---|---|---|---|---|---|
| | | *p* | *R* | *p* | *R* | *p* | *R* | *p* | *R* | *p* | *R* | *p* | *R* |
| *MD System* | Controls | 0.01* | 0.50 | 0.02 | 0.43 | 0.89 | -0.03 | 0.02 | -0.43 | 0.75 | 0.06 | 0.86 | -0.04 |
| | MS | 0.87 | -0.03 | 0.93 | 0.02 | 0.06 | 0.36 | 0.97 | 0.01 | 0.06 | 0.39 | 0.67 | 0.08 |
| | All | 0.15 | 0.19 | 0.21 | 0.17 | 0.22 | 0.16 | 0.14 | -0.20 | 0.09 | 0.23 | 0.61 | 0.07 |
| *DMN* | Controls | 0.00* | 0.51 | 0.01 | 0.47 | 0.69 | -0.08 | 0.05 | -0.37 | 0.65 | 0.09 | 0.92 | -0.02 |
| | MS | 0.64 | -0.09 | 0.90 | -0.03 | 0.06 | 0.36 | 0.86 | 0.04 | 0.07 | 0.37 | 0.68 | 0.08 |
| | All | 0.27 | 0.15 | 0.21 | 0.17 | 0.35 | 0.13 | 0.28 | -0.15 | 0.09 | 0.24 | 0.83 | 0.03 |
| *Somatosensory* | Controls | 0.03 | 0.41 | 0.03 | 0.40 | 0.44 | -0.15 | 0.00 | -0.52 | 0.98 | 0.01 | 0.63 | 0.09 |
| | MS | 0.58 | -0.11 | 0.78 | -0.06 | 0.14 | 0.29 | 0.44 | 0.15 | 0.03 | 0.45 | 0.75 | 0.06 |
| | All | 0.33 | 0.13 | 0.32 | 0.13 | 0.45 | 0.10 | 0.27 | -0.15 | 0.08 | 0.25 | 0.52 | 0.09 |

Non-parametric Spearman correlation between test scores and grey matter in the multiple demand system. P: p-value, R: Spearman-R.

* significant after Bonferroni correction for multiple comparison (*p*-value < .01).

As stated above, the results described in this paper are consistent with our previous research. Previously, we have analyzed a variety of psychological and neurological populations, including patients with frontal lobe lesions [32], Parkinson's disease [33], Frontotemporal Dementia [34], Schizophrenia [35], and Bipolar Disorder [36]. We have found that, in general, deficits in classical executive tasks can largely be explained by a loss in *g*, but deficits in multi-tasking and social cognition cannot. Our results in the MS population are compatible with our previous research. Our results are also in line with a recent study showing that, in MS patients, impairment in Theory of Mind (measured both with the "mind in the eyes" and a video test) is independent from impairment in executive functions [73].

Surprisingly, contrary to what was expected due to the common complaints of MS patients regarding multitasking and previous research [35], in the present study no deficits were found in the Hotel task of multitasking. There is a possibility that this is due to the rather small

**Table 4. Correlation between grey matter volume of each mask.**

| Mask | Group | Spearman correlation | |
|---|---|---|---|
| | | *P* | *R* |
| MD System vs DMN | Controls | <0.01* | 0.93 |
| | MS | <0.01* | 0.99 |
| | All | <0.01* | 0.96 |
| DMN vs Somatosensory | Controls | <0.01* | 0.87 |
| | MS | <0.01* | 0.93 |
| | All | <0.01* | 0.91 |
| Somatosensory vs. MD System | Controls | <0.01* | 0.89 |
| | MS | <0.01* | 0.94 |
| | All | <0.01* | 0.91 |

Non-parametric Spearman correlation between the 3 different masks of grey matter (multiple demand system, default mode network and somatosensory). P: p-value, R: Spearman-R.

* significant after bonferroni correction for multiple comparison (*p*-value < .02).

sample size of our study. Our results are also in line with a recent study that showed through cluster analysis that in MS patients impairment in Theory of Mind (measured both with the "mind in the eyes" and a video test) is independent from impairment in executive functions [73].

Regarding grey matter volumes, we searched for a link between MS cognitive deficits and the MD system [37]. When taking the control group alone, we see that Matrix Reasoning correlates with MD and DMN grey matter volume, while Faux Pas scores do not. The lack of specificity in the association between Matrix Reasoning and grey matter networks could be related to the moderate sample size of our study. Given that grey matter volumes of each network were highly correlated with one another, very large samples would be needed to have any real power to separate the specific contribution of each network. Nevertheless, the significant association between MD grey matter and the Matrix Reasoning score, together with the robust absence of association between Faux Pax and Matrix Reasoning scores, supports our previous findings regarding fluid intelligence and social cognition tests. Future and larger studies should be performed to corroborate this preliminary evidence.

Though the frontal lobes contribute to multiple cognitive functions, separating those functions has remained difficult to achieve. Taking into account the data from this paper together with our previous results on other neurological and psychiatric conditions, there seems to be a parcellation of cognitive functions based on the role of fluid intelligence. Here, as in our previous research, cognitive deficits in the set of classical executive tests were largely explained by a loss in fluid intelligence (no statistical deficit remained after the effects of fluid intelligence were partialled out). In contrast, for the social cognition test, deficits could not to be explained by a loss in fluid intelligence, since they remained after taking fluid intelligence as a covariate. Thus, we propose that deficits in classical executive tests might be explained by damage to the distributed frontoparietal MD system, likely including its white matter connections [74], although from our results we cannot rule out a possible relation to DMN or the somatosensory network. On the other hand, coherently with previous literature, deficits in social cognition may be related to impaired function in the anterior prefrontal cortex, outside the MD system (e.g. [75–78]).

Given the wide range of data published on the effect of fatigue in the neuropsychological evaluation of MS patients, we asked whether fatigue, instead of fluid intelligence, could lead to poor performance in cognitive tasks. Our results show that fatigue levels could not explain the impairments observed, while fluid intelligence losses could. This is in agreement with several studies [16–18,73,79,80] that reported no effect of fatigue on cognitive tasks. Regarding depression, it could only explain part of the impairment, specifically in the verbal fluency task, while the statistical differences for the other tests remained, supporting thus the idea that fluid intelligence, rather than depression, is strongly linked to the performance in classical executive tests.

Our data could have interesting implications for using and interpreting the performance of MS patients on classical executive tests, such as WCST, Verbal Fluency, and TMTB. Deficits detected by these tests may not be related to their cognitive content in particular, but rather to a general cognitive loss. For a comprehensive cognitive assessment, MS patients should go through fluid intelligence tests and, separately, through social cognition tests. In our view, this approach would allow healthcare providers to find potential deficits inside and outside the MD system, providing them with a more complete picture of each patient's cognitive disabilities. It should be noted that, since the present study only dealt with RRMS, the suggested approach cannot be extrapolated to other clinical forms of MS (Primary Progressive MS, Secondary Progressive MS, and Clinically-isolated Syndromes) until future studies indicate whether similar conclusions also apply to those patient populations.

Various limitations can be pointed out for the present study. Evidently the small sample size has limited our ability to draw strong conclusions in our imaging data. Further studies

need to pursue the investigation of MS cognitive deficits and fluid intelligence using imaging studies with larger sample sizes. Additionally, the fact that only patients with RRMS were included in the present study, make our findings nor generalizable to populations with other forms of the disease. Finally, although three of the most classical executive tests have been investigated in the present study, the relationship between fluid intelligence and other executive tests should be further studied, in this and in other forms of multiple sclerosis.

## Author Contributions

**Conceptualization:** Belén Goitia, Mariano Sigman, John Duncan, María Roca.

**Data curation:** Belén Goitia, Diana Bruno, Sofía Abrevaya, Vladimiro Sinay, Teresa Torralva.

**Formal analysis:** Belén Goitia, Sofía Abrevaya, Lucas Sedeño.

**Funding acquisition:** Agustín Ibáñez.

**Methodology:** Belén Goitia, Diana Bruno, Lucas Sedeño, Agustín Ibáñez, Teresa Torralva.

**Project administration:** Facundo Manes, Mariano Sigman, María Roca.

**Resources:** Agustín Ibáñez, Facundo Manes, Mariano Sigman, Vladimiro Sinay, John Duncan.

**Software:** John Duncan.

**Supervision:** Agustín Ibáñez, María Roca.

**Validation:** Vladimiro Sinay.

**Writing – original draft:** Belén Goitia, María Roca.

**Writing – review & editing:** Belén Goitia, Diana Bruno, Sofía Abrevaya, Lucas Sedeño, Agustín Ibáñez, Facundo Manes, Mariano Sigman, Vladimiro Sinay, Teresa Torralva, John Duncan, María Roca.

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
