## [Decision Letter · Decision Letter 0]

19 Dec 2019

PONE-D-19-25834

The relationship between executive functions and fluid intelligence in Multiple Sclerosis

PLOS ONE

Dear Dr. Roca,

Thank you for submitting your manuscript to PLOS ONE. After careful consideration, we feel that it has merit but does not fully meet PLOS ONE’s publication criteria as it currently stands. Therefore, we invite you to submit a revised version of the manuscript that addresses the points raised during the review process.

Both reviewers as well as myself found that your manuscript interesting but feel that it would be considerably strengthened by carefully revising several aspects. There are a number of areas where additional clarity or justifications could be made to better motivate the overall work and strengthen the findings. Please also pay extra attention to areas where the reviewers have suggested that there may be a risk of over-interpretation of some of your results. If you can strengthen these claims, then you are of course welcome to do so.

We would appreciate receiving your revised manuscript by Feb 02 2020 11:59PM. To enhance the reproducibility of your results, we recommend that if applicable you deposit your laboratory protocols in protocols.io, where a protocol can be assigned its own identifier (DOI) such that it can be cited independently in the future. For instructions see: http://journals.plos.org/plosone/s/submission-guidelines#loc-laboratory-protocols

We look forward to receiving your revised manuscript.

Kind regards,

Niels Bergsland

Academic Editor

PLOS ONE

Journal Requirements:

https://www.frontiersin.org/articles/10.3389/fnbeh.2014.00046/full

https://www.sciencedirect.com/science/article/pii/S0165178116315116?via%3Dihub

In your revision ensure you cite all your sources (including your own works), and quote or rephrase any duplicated text outside the methods section. Further consideration is dependent on these concerns being addressed.

Reviewers' comments:

Reviewer's Responses to Questions

**Comments to the Author**

1. Is the manuscript technically sound, and do the data support the conclusions?

Reviewer #1: Partly

Reviewer #2: Partly

2. Has the statistical analysis been performed appropriately and rigorously? 

Reviewer #1: Yes

Reviewer #2: Yes

3. Have the authors made all data underlying the findings in their manuscript fully available?

Reviewer #1: No

Reviewer #2: No

4. Is the manuscript presented in an intelligible fashion and written in standard English?

Reviewer #1: No

Reviewer #2: Yes

5. Review Comments to the Author

Reviewer #1: Summary:

The authors of this study aimed to examine the role of fluid intelligence (g) in executive dysfunction in patients with multiple sclerosis by using several classical and non-traditional tests of executive functioning, as well as MRI. This is an interesting topic, as executive dysfunction is common in MS, and its underlying mechanisms require further investigation. However, the manuscript has a number of issues that detract from its interesting findings.

Overall Comments:

-Overall, more care is needed in explaining the authors’ reasoning (see below). The Introduction and Discussion sections need particular improvement, including an improved discussion of the prior MS literature and how it relates specifically to this study’s scope.

-The manuscript would benefit from proofreading for typos, clarity/word usage, and grammar/punctuation. There are also some inaccurate statements and conclusions which should be addressed.

Introduction:

-The Introduction should include a more detailed explanation of concepts specifically related to this study and how they relate to MS, particularly Spearman’s g, executive functioning (e.g., what is meant by “classical executive functions tests”), multitasking, and theory of mind. What are the authors’ specific reasons for studying this topic in MS, as opposed to the other populations that are cited? How does the MD system relate to MS specifically? Why were these particular tests chosen for the study (executive functioning, multitasking, theory of mind), and have other studies found these tests to be impaired in MS? What is the reasoning behind the predictions in the authors’ hypothesis?

-Lines 45-48: Please add citation(s).

-Lines 51-54: The characterization of cognitive deficits associated with MS should be improved. MS cognitive impairments don’t “mainly include deficits in visuospatial perception and executive functions,” and this is not stated by Chiaravalloti & Deluca (2008). Rather, they cite difficulties with “complex attention, efficiency of information processing, executive functioning, processing speed, and long-term memory.”

Methods:

-Line 83: Please add a citation for the Poser and McDonald criteria that were used.

-Line 98-99: Who examined the patients and controls for comorbidities?

-Fluid intelligence (g): Is Matrix Reasoning a commonly used measure of g? If so, please cite other studies that do so. Is there a reason WAIS-III was used rather than a more updated version of the WAIS?

-WCST: Why was the Nelson’s modified version used? As the participant’s first sorting choice becomes the first correct feature, and participants are told that the rules have changed, it seems that this test is less challenging than the traditional version.

-Hotel task: It’s questionable whether this is truly a multitasking activity, as tasks were completed one at a time; rather, this seems like more a test of task-switching, goal-monitoring, and memory for intended actions. Manly (2002) referred to this as a task of “ability to monitor the time, switch between the tasks and keep track of their intentions.” Also, was there a limit on how many times the participants could consult the clock? Please also specify which version of the Hotel task was used (i.e., Manly versus Torralva).

-Faux pas: Is this the same version of the test used by Stone (1998)? Were any modifications made, such as providing the written version of the story? Did the written version of the story remain in front of the participant when they were asked questions about it?

-Were all tests administered in Spanish? If so, were all participants fluent in Spanish? Why are some executive functioning test data unavailable for some patients? How many patients completed the Faux Pas task? Did all control participants complete all executive functioning tests?

-MRI acquisition: Please explain why just a subset of patients and controls underwent MRI scanning. Was this subset matched for age, gender, and education? Why weren’t analyses performed with the other cognitive tests?

Results:

-A significant statistical difference between the two groups does not necessarily imply clinical impairment (e.g., lines 211, 214).

-Line 213: The between-group p-value for Verbal Fluency is different in the text (p=.041) than in Table 2 (p=.005); please check this.

-Line 221: What were the correlations and p-values with classical/non-classical tests and g for the MS and control groups (ie, not combined groups)? Please explain why data from patients and controls were combined for the correlations with g.

-Figure 1: Would be helpful to also include scatterplots for the non-classical tests (Hotel task and Faux Pas)

-Matrix reasoning is referred to as “g” in the initial Results section, while the other tests are referred to by their test names. It would be preferable to use the name “Matrix reasoning” rather than “g” (as is done in the Grey-matter Analysis results section), as the Matrix reasoning test is a stand-in estimate for the concept of g, rather than being g itself.

-Lines 230-231: Be careful in stating that frontal deficits were “entirely” explained by fluid intelligence – this is likely an overstatement.

-Depression is mentioned in the Introduction, and the Methods section states that depression was measured with the BDI. However, depression was not included in analyses in the Results section.

-Grey-matter analysis: This section seems incomplete and could benefit from a more detailed/clear explanation of the results. For example, please explain why patient and control data was combined in some analyses. Also, the text states that the somatosensorial mask had a significant correlation in the combined group, but this correlation does not seem to be indicated as significant in Table 3 (Lines 250-251).

Discussion:

-Be careful not to overstate or overgeneralize the conclusions. For example, “We found that MS patients show clear deficits in all classical executive tests” (Only three tests were used in this study, not “all” available executive tests. This does not imply that all MS patients have executive dysfunction, merely that there was a difference at the group level in this patient sample. Also, significant group differences do not necessarily imply clinical impairment.) Another example is the statement that “Most imaging studies regarding MS are exploratory” – this is inaccurate. The authors also state that “Cognitive deficits were entirely explained by a loss in fluid intelligence (no clinical deficit remained)”; this overstates the results and does not make sense clinically.

-As discussed above, the Hotel task does not appear to be a test of multitasking. As such, be careful about making conclusions regarding multitasking in this study.

-Why do the authors think between-group differences were non-significant for the Hotel task? Have other authors found difficulties with similar tasks in MS?

-Starting at line 283: The authors state that they aimed to search for a link between MS cognitive deficits and the MD system. However, it seems these analyses were only performed for Matrix Reasoning and Faux Pas tests; also, the results weren’t significant for the MS group (this finding is not specifically stated in the Discussion). It therefore seems a bit misleading to cite the results of the control group alone and together with the patient group, as these don’t relate to the MS findings per se.

-Lines 296-298 (“Nevertheless… social cognition test”): This statement is unclear.

-Line 303 (“support a parcellation of cognitive functions based on the role of fluid intelligence”): This statement is unclear as well; please clarify.

-Line 306: Multitasking was not included in the covariate analyses.

-The conclusions of the last paragraph seem to be overstated based on the scope and results of the study. As Matrix reasoning can also be interpreted as a test of nonabstract reasoning, reduced performance on this test may not necessarily imply a “general cognitive loss” that explains deficits on classical executive tests.

-A discussion of study limitations is largely missing; only one limitation is mentioned at the end of the manuscript. Similarly, future directions for the research should be explained further.

Reviewer #2: This manuscript provides an interesting insight into the relationship between fluid intelligence and performance on a variety of executive function tasks. Controlling for g in the MS-HC comparisons helped illustrate their main point and the additional analysis of fatigue was also appreciated as a useful comparison.

Introduction: The introduction is mostly focused on cognition at large and only 1-2 sentences are dedicated to introducing your background work, relating g to executive function tasks. This is merely a recommendation, however I would suggest cutting back some of the opening text on cognition at large so that there is more space to provide readers more immediately relevant background.

Analysis/Results:

- Page 10 & 11. Regarding the correlations between g and other tests. Here you provide "average within-group correlations", yet in the MRI section, you instead provide correlations which reflect the correlation in either MS, HCs, or both. Please be consistent. It would probably be best to do the same here and provide all 3 correlations for these test values with g. Please also provide p-values for these correlations in Table 2.

- Page 11: The statement "This suggests that, for classical executive tasks, frontal deficits were entirely explained by fluid intelligence" is not fully supported by the results. The results of the ANCOVA at least suggest that g and the executive function tasks have overlapping variance. That said, we can not conclude the frontal deficits are ENTIRELY explained by g. I would suggest pulling back on this conclusion and perhaps moving the interpretation to the discussion section.

- Page 13: Table 3 results show that g correlates with MD, DMN, and somatosensory network GM volume. Please add a star to the p-value for the somatosensory, as this indicates statistical significance (P<0.02). Also, these results show that correlation with g is not unique to the MD and DMN, since the correlation appears for somatosensory network as well. To make the intended point, the author should see if these correlations with g remain after controlling for whole brain gray matter volume. That way, if the significance of the local correlations remain, the authors will know that this correlation is locally specific. The authors may just be picking up on the expected correlation between gray matter volume at large and cognition at large rather than for their intended hypothesized correlation between g and volume of MD system specifically. As an aside, perhaps change "raw matrix" in Table 3 to "g" so is consistent with rest of manuscript.

Discussion:

- page 15. On the bottom, the authors again assert that the cognitive deficits are ENTIRELY explained by g. This is not supported by the ANCOVA results. Please provide a more conservative interpretation of these results.

- Page 16. The authors say "we propose that deficits in classical executive tasks might be explained by damage to the distributed frontoparietal MD system". As they stand, the results don't provide support for this conclusion uniquely. From the results presented, one could also conclude that DMN and somatosensory are equally related.

6. PLOS authors have the option to publish the peer review history of their article (what does this mean?). If published, this will include your full peer review and any attached files.

Reviewer #1: No

Reviewer #2: No

---

## [Author Response · Author response to Decision Letter 0]

21 Feb 2020

Reviewer #1: Summary:

The authors of this study aimed to examine the role of fluid intelligence (g) in executive dysfunction in patients with multiple sclerosis by using several classical and non-traditional tests of executive functioning, as well as MRI. This is an interesting topic, as executive dysfunction is common in MS, and its underlying mechanisms require further investigation. However, the manuscript has a number of issues that detract from its interesting findings.

Overall Comments:

-Overall, more care is needed in explaining the authors’ reasoning (see below). The Introduction and Discussion sections need particular improvement, including an improved discussion of the prior MS literature and how it relates specifically to this study’s scope.

We thank the reviewer for this comment. We have re-written both the introduction and discussion according to the reviewer’s suggestions and we hope we have addressed the reviewer’s concerns.

-The manuscript would benefit from proofreading for typos, clarity/word usage, and grammar/punctuation. There are also some inaccurate statements and conclusions which should be addressed.

The manuscript has been checked for typos, word usage, grammar, and punctuation, as requested. 

Introduction:

-The Introduction should include a more detailed explanation of concepts specifically related to this study and how they relate to MS, particularly Spearman’s g, executive functioning (e.g., what is meant by “classical executive functions tests”), multitasking, and theory of mind. What are the authors’ specific reasons for studying this topic in MS, as opposed to the other populations that are cited? How does the MD system relate to MS specifically? Why were these particular tests chosen for the study (executive functioning, multitasking, theory of mind), and have other studies found these tests to be impaired in MS? What is the reasoning behind the predictions in the authors’ hypothesis?

We are very grateful for this comment and we have now re-written the introduction including the reviewer’s suggestions. We believe this has definitively strengthened our manuscript.

-Lines 45-48: Please add citation(s).

Added as requested. 

-Lines 51-54: The characterization of cognitive deficits associated with MS should be improved. MS cognitive impairments don’t “mainly include deficits in visuospatial perception and executive functions,” and this is not stated by Chiaravalloti & Deluca (2008). Rather, they cite difficulties with “complex attention, efficiency of information processing, executive functioning, processing speed, and long-term memory.”

We agree that the description of MS cognitive deficits was quite vague. We have now included a more detailed description of the cognitive deficits found in MS. 

Methods:

-Line 83: Please add a citation for the Poser and McDonald criteria that were used.

Added as requested. 

-Line 98-99: Who examined the patients and controls for comorbidities?

Patients were initially examined for comorbidities by the physician who selected the subjects for the study (VS). Controls were selected from a panel of control subjects in which they initially completed a questionnaire where the presence or absence of relevant comorbidities was assessed. Then, the neuropsychologist who assessed both patients and controls performed a short interview where subjects’ medical history was discussed. Only those subjects who fit the study criteria were included in the present investigation.

-Fluid intelligence (g): Is Matrix reasoning a commonly used measure of g? If so, please cite other studies that do so. 

The best tests of fluid intelligence, this is to say the most predictive of a general ability to do well are those calling for novel problem-solving with simple visual or other materials (Cattell, 1971; Carroll, 1993). Widely used examples are Raven’s Matrices (Raven, 1938; Raven et al., 1988) and Cattell’s Culture Fair (Institute for Personality and Ability Testing, 1973). 

-Is there a reason WAIS-III was used rather than a more updated version of the WAIS?

Subjects were tested in Argentina at the time the WAIS III was the available version. 

-WCST: Why was the Nelson’s modified version used? As the participant’s first sorting choice becomes the first correct feature, and participants are told that the rules have changed, it seems that this test is less challenging than the traditional version.

We agree that the traditional version of the test is commonly used. However, the Nelson version is also used in many studies and there is evidence that this version correlates highly with the long form (Spreen & Strauss, A compendium of neuropsychological tests. Oxford University Press)

-Hotel task: It’s questionable whether this is truly a multitasking activity, as tasks were completed one at a time; rather, this seems like more a test of task-switching, goal-monitoring, and memory for intended actions. Manly (2002) referred to this as a task of “ability to monitor the time, switch between the tasks and keep track of their intentions.” Also, was there a limit on how many times the participants could consult the clock? Please also specify which version of the Hotel task was used (i.e., Manly versus Torralva).

We have described the hotel task as a multitasking test in order to be consistent with both the literature describing performance on this test in MS and other pathologies and the literature describing the relationship between this task and fluid intelligence. The test is supposed to assess multitasking since several goals and actions have to be addressed simultaneously (eg. doing a certain task while remembering to open or close the garage doors). However, we definitely agree with the reviewer that the hotel task tackles multiple and complex processes and we have included a better description of the test to address the reviewers concern . 

The Torralva´s version of the test was used in which there is no limit as to how many times the participants could consult the clock.

-Faux pas: Is this the same version of the test used by Stone (1998)? Were any modifications made, such as providing the written version of the story? Did the written version of the story remain in front of the participant when they were asked questions about it?

Again, the Torralva´s Argentinean adaptation of the test was used. A written version of the story was provided which was available to the subject when the questions were asked. The participants were allowed to read back the story as many times as needed. We have now described this procedure in further detail in the method section. 

-Were all tests administered in Spanish? If so, were all participants fluent in Spanish? Why are some executive functioning test data unavailable for some patients? How many patients completed the Faux Pas task? Did all control participants complete all executive functioning tests?

The tests were administered in Spanish, since all the subjects were Argentinean and Spanish native speakers. The unavailability of part of the data is due to the fact that the present investigation was part of an ongoing investigation, during which, at a certain point, this test was included. All subjects completed the Faux Pas and all controls participants completed all executive functioning tests.

-MRI acquisition: Please explain why just a subset of patients and controls underwent MRI scanning. Was this subset matched for age, gender, and education? Why weren’t analyses performed with the other cognitive tests?

Unfortunately, some participants were unavailable to perform the MRI sessions. The subset of subjects in whom MRI was performed did not differed from those in which MRI was not performed. The sub-group analysed was matched in age, gender and education (see Table A in "Response to Reviewers"). On the other hand, following our principal hypothesis we only tested correlation with the cognitive test of interest. But in Table 3, we now report all of them.

To clarify the results, we have performed also the correlation with the other cognitive test, see modified Table 3 in the manuscript.

Results:

-A significant statistical difference between the two groups does not necessarily imply clinical impairment (e.g., lines 211, 214).

We can see how from the previous way in which our manuscript was written the reviewer might have been led to think that we were implying clinical impairment. However, that was never our intention. We believe that, after tempering several of our claims, the reviewer will find that there is no mention of a clinical impairment in our manuscript. 

-Line 213: The between-group p-value for Verbal Fluency is different in the text (p=.041) than in Table 2 (p=.005); please check this.

We thank the reviewer for pointing out this mistake. We have now corrected the table.

-Line 221: What were the correlations and p-values with classical/non-classical tests and g for the MS and control groups (ie, not combined groups)? Please explain why data from patients and controls were combined for the correlations with g.

We thank the reviewer for drawing our attention to this point. Please find the information requested in Teble B in "Response to Reviewers".

-Figure 1: Would be helpful to also include scatterplots for the non-classical tests (Hotel task and Faux Pas). 

We have now changed the figure and included a new one with all the scatterplots requested. 

-Matrix reasoning is referred to as “g” in the initial Results section, while the other tests are referred to by their test names. It would be preferable to use the name “Matrix reasoning” rather than “g” (as is done in the Grey-matter Analysis results section), as the Matrix reasoning test is a stand-in estimate for the concept of g, rather than being g itself.

We agree with the reviewer point and we have changed the manuscript as requested. 

-Lines 230-231: Be careful in stating that frontal deficits were “entirely” explained by fluid intelligence – this is likely an overstatement.

We have now tempered the statement and we hope we have addressed the reviewer’s fair concern. 

-Depression is mentioned in the Introduction, and the Methods section states that depression was measured with the BDI. However, depression was not included in analyses in the Results section.

We thank the reviewer for drawing our attention to this point. We have now added the information requested in table 2. 

-Grey-matter analysis: This section seems incomplete and could benefit from a more detailed/clear explanation of the results. For example, please explain why patient and control data was combined in some analyses. Also, the text states that the somatosensorial mask had a significant correlation in the combined group, but this correlation does not seem to be indicated as significant in Table 3 (Lines 250-251).

We have added an explanation of the rationality of combining patient and control data in the Method section

Discussion:

-Be careful not to overstate or overgeneralize the conclusions. For example, “We found that MS patients show clear deficits in all classical executive tests” (Only three tests were used in this study, not “all” available executive tests. This does not imply that all MS patients have executive dysfunction, merely that there was a difference at the group level in this patient sample. Also, significant group differences do not necessarily imply clinical impairment.) Another example is the statement that “Most imaging studies regarding MS are exploratory” – this is inaccurate. The authors also state that “Cognitive deficits were entirely explained by a loss in fluid intelligence (no clinical deficit remained)”; this overstates the results and does not make sense clinically.

We thank the reviewer for this comment since we now realize that we might have been too enthusiastic in our conclusions. We have now tempered our statements, hoping again we have addressed the reviewer’s concerns. 

-As discussed above, the Hotel task does not appear to be a test of multitasking. As such, be careful about making conclusions regarding multitasking in this study.

We have already addressed this point and we have re written our conclusion regarding this test. 

-Why do the authors think between-group differences were non-significant for the Hotel task? Have other authors found difficulties with similar tasks in MS?

A previous study found significant differences between MS and controls. We believe that the explanation for our lack of significance could be related to small sample size. We have now included this reasoning in the discussion of our results.

-Starting at line 283: The authors state that they aimed to search for a link between MS cognitive deficits and the MD system. However, it seems these analyses were only performed for Matrix Reasoning and Faux Pas tests; also, the results weren’t significant for the MS group (this finding is not specifically stated in the Discussion). It therefore seems a bit misleading to cite the results of the control group alone and together with the patient group, as these don’t relate to the MS findings per se.

We thank the reviewer for this comment. We have now clarify this point in the discussion preventing misleading interpretations. 

-Lines 296-298 (“Nevertheless… social cognition test”): This statement is unclear.

We have now re-written the statement in order to make it clearer. 

-Line 303 (“support a parcellation of cognitive functions based on the role of fluid intelligence”): This statement is unclear as well; please clarify.

We have now re-written the statement in order to make it clearer. 

-Line 306: Multitasking was not included in the covariate analyses.

We did not include the hotel test in the covariance analyses since there were not significant differences in the first place. 

-The conclusions of the last paragraph seem to be overstated based on the scope and results of the study. As Matrix reasoning can also be interpreted as a test of non-abstract reasoning, reduced performance on this test may not necessarily imply a “general cognitive loss” that explains deficits on classical executive tests.

As we stated above, we have now tempered our claims hoping we have addressed the reviewer’s concern. 

-A discussion of study limitations is largely missing; only one limitation is mentioned at the end of the manuscript. Similarly, future directions for the research should be explained further.

We have described the study limitations throughout the discussion of the results. 

Reviewer #2: 

This manuscript provides an interesting insight into the relationship between fluid intelligence and performance on a variety of executive function tasks. Controlling for g in the MS-HC comparisons helped illustrate their main point and the additional analysis of fatigue was also appreciated as a useful comparison

Introduction: The introduction is mostly focused on cognition at large and only 1-2 sentences are dedicated to introducing your background work, relating g to executive function tasks. This is merely a recommendation, however I would suggest cutting back some of the opening text on cognition at large so that there is more space to provide readers more immediately relevant background.

We thank the reviewer for this point and we have now re-written our introduction accordingly. 

Analysis/Results:

- Page 10 & 11. Regarding the correlations between g and other tests. Here you provide "average within-group correlations", yet in the MRI section, you instead provide correlations which reflect the correlation in either MS, HCs, or both. Please be consistent. It would probably be best to do the same here and provide all 3 correlations for these test values with g. Please also provide p-values for these correlations in Table 2.

We thank the reviewer for the comment. When data is not homogeneous (as in this case where we have two groups), there are three different correlations: 1) correlation, ignoring the existence of groups altogether; 2) correlation within the groups; and 3) correlation across the groups [1]. Taking this into account we performed a conjoint (for point 1) and separated by group correlation analysis (for point 3) between the grey matter values and cognitive test scores. We consider addressing point 2 is unnecessary, since we seek to contrast an overall correlation. Furthermore, we have previously done this type of analysis with the same population [2] and other neurodegenerative diseases [3].

Regarding whole brain volume correction, we tested first whether there were differences in total intracranial volume (TIV) and total grey-matter volume (TGMV) between the two groups, and whether these values correlated with the volume of the mask used. For both cases no differences or associations were found (see Tables C and D in "Response to Reviewers"). Other studies have also not corrected mask volume by total volume [4].

- Page 11: The statement "This suggests that, for classical executive tasks, frontal deficits were entirely explained by fluid intelligence" is not fully supported by the results. The results of the ANCOVA at least suggest that g and the executive function tasks have overlapping variance. That said, we can not conclude the frontal deficits are ENTIRELY explained by g. I would suggest pulling back on this conclusion and perhaps moving the interpretation to the discussion section.

We thank the reviewer for this comment and we have now tempered our claims throughout the text. 

- Page 13: Table 3 results show that g correlates with MD, DMN, and somatosensory network GM volume. Please add a star to the p-value for the somatosensory, as this indicates statistical significance (P<0.02). Also, these results show that correlation with g is not unique to the MD and DMN, since the correlation appears for somatosensory network as well. To make the intended point, the author should see if these correlations with g remain after controlling for whole brain gray matter volume. That way, if the significance of the local correlations remain, the authors will know that this correlation is locally specific. The authors may just be picking up on the expected correlation between gray matter volume at large and cognition at large rather than for their intended hypothesized correlation between g and volume of MD system specifically. As an aside, perhaps change "raw matrix" in Table 3 to "g" so is consistent with rest of manuscript.

We thank the reviewer for the suggestion, and the corrections have been made. Regarding the whole brain grey matter volume correction, we have answered it in a previous question. 

Discussion:

- page 15. On the bottom, the authors again assert that the cognitive deficits are ENTIRELY explained by g. This is not supported by the ANCOVA results. Please provide a more conservative interpretation of these results.

As we said above we thank the reviewer for this comment and we have now tempered our claims throughout the text. 

- Page 16. The authors say "we propose that deficits in classical executive tasks might be explained by damage to the distributed frontoparietal MD system". As they stand, the results don't provide support for this conclusion uniquely. From the results presented, one could also conclude that DMN and somatosensory are equally related.

We thank the reviewer for this comment. We have now clarify this point in the discussion preventing misleading interpretations. 

References

 1. Marzban, C., et al., Within-group and between-group correlation: Illustration on non-invasive estimation of intracranial pressure. viewed nd, from http://faculty. washington. edu/marzban/within_ between_simple. Pdf, 201.3

 2. Gonzalez Campo, C., et al., Fatigue in multiple sclerosis is associated with multimodal interoceptive abnormalities. Mult Scler, 2019: p. 1352458519888881.

 3. Garcia-Cordero, I., et al., Feeling, learning from and being aware of inner states: interoceptive dimensions in neurodegeneration and stroke. Philos Trans R Soc Lond B Biol Sci, 2016. 371(1708).

 4. Melloni, M., et al., Cortical dynamics and subcortical signatures of motor-language coupling in Parkinson's disease. Sci Rep, 2015. 5: p. 11899.

---

## [Decision Letter · Decision Letter 1]

3 Apr 2020

The relationship between executive functions and fluid intelligence in Multiple Sclerosis

PONE-D-19-25834R1

Dear Dr. Roca,

We are pleased to inform you that your manuscript has been judged scientifically suitable for publication and will be formally accepted for publication once it complies with all outstanding technical requirements.

With kind regards,

Niels Bergsland

Academic Editor

PLOS ONE

Additional Editor Comments (optional):

Reviewers' comments:

Reviewer's Responses to Questions

**Comments to the Author**

1. If the authors have adequately addressed your comments raised in a previous round of review and you feel that this manuscript is now acceptable for publication, you may indicate that here to bypass the “Comments to the Author” section, enter your conflict of interest statement in the “Confidential to Editor” section, and submit your "Accept" recommendation.

Reviewer #1: All comments have been addressed

2. Is the manuscript technically sound, and do the data support the conclusions?

Reviewer #1: (No Response)

3. Has the statistical analysis been performed appropriately and rigorously? 

Reviewer #1: (No Response)

4. Have the authors made all data underlying the findings in their manuscript fully available?

Reviewer #1: (No Response)

5. Is the manuscript presented in an intelligible fashion and written in standard English?

Reviewer #1: (No Response)

6. Review Comments to the Author

Reviewer #1: (No Response)

7. PLOS authors have the option to publish the peer review history of their article (what does this mean?). If published, this will include your full peer review and any attached files.

Reviewer #1: No

---

## [Editor Report · Acceptance letter]

8 Apr 2020

PONE-D-19-25834R1 

The relationship between executive functions and fluid intelligence in Multiple Sclerosis 

Dear Dr. Roca:

I am pleased to inform you that your manuscript has been deemed suitable for publication in PLOS ONE. Congratulations! Your manuscript is now with our production department. 

With kind regards,

on behalf of

Dr. Niels Bergsland 

Academic Editor

PLOS ONE